The origin of widespread species in a poor dispersing lineage (diving beetle genus Deronectes)

García-Vázquez David
Ribera Ignacio ignacio.ribera@ibe.upf-csic.es
Institute of Evolutionary Biology (CSIC-Universitat Pompeu Fabra) , Barcelona , Spain
Riutort Marta
Electronic publication date: 2016 Sep 27
Publication date: 2016
Volume: 4
Electronic Location ID: e2514
Received 2016 Jun 23; Accepted 2016 Sep 1
Copyright: ©2016 García-Vázquez and Ribera
Copyright year: 2016
Copyright holder: García-Vázquez and Ribera
License: This is an open access article distributed under the terms of the Creative Commons Attribution License, which permits unrestricted use, distribution, reproduction and adaptation in any medium and for any purpose provided that it is properly attributed. For attribution, the original author(s), title, publication source (PeerJ) and either DOI or URL of the article must be cited.
License URL: https://creativecommons.org/licenses/by/4.0/

Keywords: Dispersion, Glacial refugia, Mediterranean peninsulas, Range expansion, Pleistocene glaciations

Funding: Government of Spain Secretaria d’Universitats i Recerca del Departament d’Economia i Coneixement de la Generalitat de Catalunya SGR1532 DG-V had a FPI PhD grant from the Government of Spain. This work was partially funded by projects CGL2010-15755 and CGL2013-48950-C2-1-P to IR, and the “Secretaria d’Universitats i Recerca del Departament d’Economia i Coneixement de la Generalitat de Catalunya” (project SGR1532). The funders had no role in study design, data collection and analysis, decision to publish, or preparation of the manuscript.

==============================
In most lineages, most species have restricted geographic ranges, with only few reaching widespread distributions. How these widespread species reached their current ranges is an intriguing biogeographic and evolutionary question, especially in groups known to be poor dispersers. We reconstructed the biogeographic and temporal origin of the widespread species in a lineage with particularly poor dispersal capabilities, the diving beetle genus Deronectes (Dytiscidae). Most of the ca. 60 described species of Deronectes have narrow ranges in the Mediterranean area, with only four species with widespread European distributions. We sequenced four mitochondrial and two nuclear genes of 297 specimens of 109 different populations covering the entire distribution of the four lineages of Deronectes, including widespread species. Using Bayesian probabilities with an a priori evolutionary rate, we performed (1) a global phylogeny/phylogeography to estimate the relationships of the main lineages within each group and root them, and (2) demographic analyses of the best population coalescent model for each species group, including a reconstruction of the geographical history estimated from the distribution of the sampled localities. We also selected 56 specimens to test for the presence of Wolbachia, a maternally transmitted parasite that can alter the patterns of mtDNA variability. All species of the four studied groups originated in the southern Mediterranean peninsulas and were estimated to be of Pleistocene origin. In three of the four widespread species, the central and northern European populations were nested within those in the northern areas of the Anatolian, Balkan and Iberian peninsulas respectively, suggesting a range expansion at the edge of the southern refugia. In the Mediterranean peninsulas the widespread European species were replaced by vicariant taxa of recent origin. The fourth species (D. moestus) was proven to be a composite of unrecognised lineages with more restricted distributions around the Western and Central Mediterranean. The analysis of Wolbachia showed a high prevalence of infection among Deronectes, especially in the D. aubei group, where all sequenced populations were infected with the only exception of the Cantabrian Mountains, the westernmost area of distribution of the lineage. In this group there was a phylogenetic incongruence between the mitochondrial and the nuclear sequence, although no clear pattern links this discordance to the Wolbachia infection. Our results suggest that, in different glacial cycles, populations that happened to be at the edge of the newly deglaciated areas took advantage of the optimal ecological conditions to expand their ranges to central and northern Europe. Once this favourable ecological window ended populations become isolated, resulting in the presence of closely related but distinct species in the Mediterranean peninsulas.

Introduction

The geographic range of the species is a fundamental trait in ecology and biogeography (Brown & Lomolino, 1998). There have been many hypotheses put forward by ecologists, biogeographers or evolutionary biologists to explain geographic ranges (see e.g., Brown, 1995 or Gaston, 2003 for reviews), such as differences in niche breadth, body size, population abundance, environmental variability, colonization and extinction dynamics, and dispersal ability among others (Stevens, 1989; Brown, 1995; Gaston, 2003), but there is still a lack of understanding of the evolutionary dynamics of geographical ranges. Thus, although the ecological and biogeographic context in which species evolve are known to determine their range sizes (Böhning-Gaese et al., 2006), there are multiple cases of closely related species, sharing a common phylogenetic history and with a similar biology and ecology, that have nonetheless extreme differences in the size of their geographical range (Lester et al., 2007).

In freshwater species, one of the most general and robust associations is that of habitat stability and geographic range (Ribera, 2008; Dijkstra, Monaghan & Pauls, 2014). Species living in standing waters tend to have better dispersal abilities than species living in running waters, due to the shorter geological duration of their habitats. Running water species have on average smaller geographic ranges and a lower gene flow between populations (Ribera, 2008; Abellán, Millán & Ribera, 2009). Range size is also strongly correlated with extinction probability (Jablonski, 1987), and high probabilities of extinction have been linked with narrow geographical ranges in several taxa (see e.g., Hansen, 1980 for marine invertebrates, Purvis et al., 2000 for mammals, or Rosenfield, 2002 for freshwater fishes). This raises the question of how lineages with a predominance of running water species, and thus on average smaller geographic ranges, can persist over long evolutionary periods. It must be noted here that while among standing water species the frequency of species with small ranges is expected to be low, within running water lineages it is not rare to find widespread species (Ribera & Vogler, 2000). This is likely due to the asymmetry in the habitat constraints: standing water species have to migrate when their habitat disappears, but the higher stability of running water habitats means that species in them do not need to disperse. Nevertheless, they tend to lose dispersal capabilities, likely due to their associated cost (see Ribera, 2008 for a review), although in some circumstances running water species seem to be able to disperse and reach widespread distributions. These few widespread species may be of disproportionate importance, as potential sources of new species (“diversity pumps,” Ribera et al., 2011), but there is very few data on what makes a species in a clade of poor dispersers able to expand its range, or in what circumstances.

A common pattern among many lineages with abundance of species with restricted geographic ranges within the Mediterranean area is the presence of one (or few) species with a widespread distribution including parts of central and northern Europe. These northern populations should have a recent origin, as most of central and northern Europe was glaciated or with permafrost during the Last Glacial Maximum, when the European ice sheet extended north of 52°N and the permafrost north of 47°N (Dawson, 1992). It is thus clear that some species were able to expand their ranges in a short time, as they could not have survived on the ice sheet.

We study the origin and phylogeography of the European widespread species in one of these running water lineages, the diving beetle genus Deronectes Sharp (family Dytiscidae). With c. 60 described species (Nilsson & Hájek, 2015), Deronectes is the largest clade of Palaearctic diving beetle entirely confined to running waters. It has a predominantly Mediterranean distribution, ranging from North Africa and the Iberian Peninsula over most parts of Europe and the Middle East, with some species reaching central Asia (Nilsson, 2001). The genus has many narrow range Mediterranean species, including some island endemics, but also a few species widespread in central and northern Europe, showing their potential for range expansion. A recent molecular phylogeny of the genus (García-Vázquez et al., 2016) supported the existence of two main lineages, mostly corresponding to species with a western or eastern distribution. The widespread European species were also shown to be of Pleistocene origin, in agreement with other works with the genus (Ribera, Barraclough & Vogler, 2001; Ribera, 2003; Ribera & Vogler, 2004; Abellán & Ribera, 2011).

Our main objective here is to reconstruct in detail the temporal and geographic origin of the widespread European species of Deronectes, using molecular data from a comprehensive sampling of them and their closest relatives, covering their entire distributions.

Material and Methods

Taxonomic background

Four species of Deronectes have widespread European distributions. Three of them are found in central and northern Europe, i.e., areas strongly affected by the last glacial cycle: (1) D. aubei (Mulsant), (2) D. latus(Stephens) and (3) D. platynotus (Germar). The fourth, D. moestus (Fairmaire), is distributed along the western Mediterranean from the Maghreb to the Balkans, although with high intraspecific variation, according to the preliminary results of García-Vázquez et al. (2016). All four belong to different species groups within the genus, as defined by García-Vázquez et al. (2016).

(1) The D. aubei group includes three species, one of them divided in two subspecies. The widespread D. aubei includes D. a. aubei from the Alps and surrounding areas, reaching southern Germany (Black forest) in the north, and D. a. sanfilippoi Fery & Brancucci distributed from the Cantabrian mountains in NW Spain to southern France (Pyrenees). The other two species are D. semirufus (Germar), from the Alps Maritims to Sicily, and D. delarouzei (Jacquelin du Val) in the Pyrenees (Fery & Brancucci, 1997) (Fig. 1). García-Vázquez et al. (2016) found incongruence between mitochondrial and nuclear data within the D. aubei group. According to the mitochondrial sequence, the two subspecies of D. aubei were recovered as paraphyletic and respectively sisters to the geographically closest species of the group: one clade was distributed west of the Rhone river, including D. a. sanfilippoi and D. delarouzei, and another east of the Rhone, with D. a. aubei and D. semirufus. On the contrary, the nuclear sequence recovered a monophyletic D. aubei as sister to the other two species of the group (D. delarouzei and D. semirufus).

Figure 1 Distribution maps of the studied species.

Known distribution of the species of the four studied groups of Deronectes. White circles, sampled localities.

(2) The D. latus group, within the eastern Mediterranean lineage of Deronectes, includes four species. Deronectes latus is the most widespread species of the genus, occupying large areas of Europe north of the Pyrenees and the Apennines, reaching the British Isles and Scandinavia. On the contrary, the other three species have restricted distributions in the southern peninsulas: D. toledoi Fery, Erman & Hosseinie in northeastern Turkey; D. angusi Fery & Brancucci in northwestern Spain; and D. angelinii Fery & Brancucci, the only Italian endemic of the genus, from the Apennines to Sicily, including the island of Elba (see Fig. 1 for the distribution of each species). The entire group was reconstructed as having an origin in the Anatolian peninsula, from were expanded first to Italy and the Balkan Peninsula and subsequently to the Iberian Peninsula and central and northern Europe (García-Vázquez et al., 2016).

(3) The D. platynotus group, in the western clade of Deronectes, comprises two species, with two subspecies each. Deronectes platynotus includes the widespread D. p. platynotus, distributed from the Balkans and northern Greece to central Europe, and the subspecies D. p. mazzoldi Fery & Brancucci, from northern Greece. The second species, the NW Iberian endemic D. costipennis Brancucci, includes D. c. costipennis from Portugal and the extreme NW Iberia, and D. c. gignouxi Fery & Brancucci, 1997, from the Cantabrian region (Fig. 1).

(4) The D. moestus group includes the fourth of the widespread European species of the genus, D. moestus, distributed from the Maghreb and the Iberian Peninsula to the Balkans through southern France, Italy and Sicily, together with other western Mediterranean species. According to the results of García-Vázquez et al. (2016) D. moestus as currently understood is paraphyletic, and includes the Mallorcan endemic D. brannani (Schauffus). The two recognised subspecies of D. moestus, D. m. moestus from Corsica and Sardinia and D. m. inconspectus (Leprieur) from the continent, were not recovered as respectively monophyletic. Here we studied the species D. moestus plus D. brannani, what we call the D. moestus complex (Fig. 1). The species D. perrinae Fery & Brancucci, from Algeria and Tunisia, is morphologically very close to D. moestus, and probably should be included in this complex, although no specimens could be obtained for study.

Taxon sampling

We studied a total of 296 specimens of 109 populations (with a maximum of 5 specimens per population) of all species of the four studied lineages of Deronectes, of which 23 specimens were previously used by García-Vázquez et al. (2016) (Table S1). We aimed to cover the full geographic range of the studied species, in particular potential cryptic refuge areas (in the sense of Homburg et al., 2013) at the margins of the Alps and neighbouring mountain ranges (such as the Black Forest), or areas not included in previous works (e.g., Sicily or French Massif Central for the species of the D. aubei group).

DNA extraction and sequencing

Specimens were collected and preserved in absolute ethanol directly in the field. We extracted the DNA non-destructively with commercial kits (mostly “DNeasy Tissue Kit”; Qiagen GmbH, Hilden, Germany, and “Charge Switch gDNA Tissue Mini Kit”; Invitrogen, Carlsbad, CA, USA) following the manufacturer’s instructions. Specimens and DNA extractions are kept in the collections of the Institut de Biología Evolutiva, Barcelona (IBE), Museo Nacional de Ciencias Naturales, Madrid (MNCN) and Natural History Museum, London (NHM).

We obtained seven gene fragments from six different genes (four mitochondrial and two nuclear) in five different amplification reactions (see Table S2 for primers and typical sequencing reactions): (1) 5′ end of the Cytochrome Oxidase Subunit 1 gene (the barcode fragment, Hebert, Ratnasingham & De Waard, 2003, COI-5′); (2) 3′ end of Cytochrome Oxidase Subunit 1 (COI-3′); (3) 5′ end of 16S rRNA plus tRNA transfer of Leucine plus 5′ end of NADH subunit 1 (nad1) (16S); and internal fragments of the nuclear genes (4) Histone 3 (H3) and (5) Wingless (Wg). The nuclear markers were only used in representative specimens according to geographical (one specimen per population) and topological criteria. For each amplification reaction, we obtained both forward and reverse sequences. In some specimens, and due to difficulties of amplification, we used internal primers for the fragment COI-3′, obtaining two fragments of 400 bp each. The obtained products were purified by standard ethanol precipitation and sent to external facilities for sequencing after purification (LMU Genomics Service Unit, Martinsried, Germany). DNA sequences were assembled and edited using the GENEIOUS 6 software (Biomatters Ltd, Auckland, New Zealand). New Deronectes sequences (a total of 791) have been deposited in GenBank with accession numbers LT601818 –LT602609, and Wolbachia sequences (23) with numbers LT602610 –LT602633 (Table S1).

Phylogenetic and phylogeographic analyses

Edited sequences were aligned with MAFFT v.6 using the G-INS algorithm and default values for other parameters (Katoh & Toh, 2008). We employed six partitions corresponding to the gene fragments COI-5′, COI-3′, 16S rRNA+tRNA-Leu, nad1, H3 and Wg, and used Partition Finder 1.1.1 (Lanfear et al., 2012) to estimate the best partitioning scheme (each gene separately or combining genes in a partition by codons) and the evolutionary model that best fitted the data for each partition separately, using the AIC (Akaike Information Criterion) as selection criteria. We performed two types of analyses using different datasets: (1) a global phylogeny/phylogeography to estimate the relationships of the main lineages within each group and root them; and (2) demographic analyses of the best population coalescent model for each group, including a reconstruction of the geographical history estimated from the distribution of the sampled localities.

For the global phylogeny/phylogeography we first selected one specimen per population to estimate the general topology of the four lineages within the wider genus Deronectes, employing Bayesian methods implemented in BEAST 1.8 (Drummond & Rambaut, 2007), using as outgroups 18 species of Deronectes in other species groups. We rooted the trees in the separation between the western and eastern clades of the genus, following García-Vázquez et al., (2016). For the analysis, we implemented the closer available evolutionary model to those selected by Partition Finder and the best partitioning scheme, including both mitochondrial and nuclear markers. As there are no fossils or unambiguous biogeographic events that could be used to calibrate the phylogeny of Deronectes (García-Vázquez et al., 2016), we applied an a-priori rate of 0.013 substitutions/site/MY (standard deviation 0.002) for protein coding genes and 0.0016 substitutions/site/MY (standard deviation 0.0002) for 16S, obtained in a related group (family Carabidae) for the same mitochondrial genes (Andújar, Serrano & Gómez-Zurita, 2012). As there are no reliable estimations of the evolutionary rate of the nuclear genes, we applied an uniform flat prior (from 0 to infinite). We used an uncorrelated lognormal relaxed clock, a Yule speciation model and executed two independent analyses with the same settings, running 100 million generations (saving trees every 5,000) or until they converged and the number of trees was sufficient according to Effective Sample Size (ESS) values, as measured with TRACER v1.6 (Drummond & Rambaut, 2007). The majority rule consensus tree of the two runs was compiled with Tree Annotator v1.8 (Drummond & Rambaut, 2007). To test for potential discordances among the different markers, we also analysed separately the mitochondrial and nuclear data. We applied the same settings as the combined analysis but, to estimate the divergence dates among species, we established as a prior for the root of the tree a normal distribution with average 14 Ma and a standard deviation of 0.1, according to the results obtained in García-Vázquez et al. (2016).

For the demographic analyses we used all sequenced specimens from all populations to estimate, for each of the four groups separately, the demographic model (constant, logistic, expansion or exponential) that best fitted the data. Previously, we tested the adjustment to a strict molecular clock and used the best clock model in the coalescent analyses. We used the mitochondrial sequence only and no outgroups, with a GTR+I+G evolutionary model and the same settings used in the analysis of the topology. For model selection of the molecular clock and coalescent analyses, we used the modified Akaike information criterion (AICM) with moments estimator (Baele et al., 2012), as implemented in TRACER v1.6, with 1,000 bootstrap replicates. We also computed Bayesian skyline plots for each group to reconstruct the variation of the effective population sizes with time.

Using the best clock and demographic models as selected above, and after excluding specimens from the same locality with repeated haplotypes, we reconstructed the phylogeographic history of each group in BEAST using the coordinates of each locality as a quantitative trait (Lemey et al., 2009). We excluded outgroups, but included the nuclear markers in the analyses. We overlaid the reconstructed coordinates of the nodes of the phylogeography of each group, as obtained with BEAST and SPREAD v1.0.6 (Bielejec et al., 2011), in Google Earth (http://earth.google.com), allowing the reconstruction through time of the history of each lineage, from the origin until the current distribution.

Wolbachia detection

The incongruence between the mitochondrial and the nuclear data in the D. aubei group noted by García-Vázquez et al. (2016) raised the possibility that specimens could be infected with Wolbachia, a maternally transmitted parasite that can alter the patterns of mtDNA variability (Jiggins, 2003). We thus tested for the presence of Wolbachia in 56 selected specimens (see Table S1) of the four studied groups of Deronectes, focusing on the species of the D. aubei group.

We used specific primers for the wsp gene, amplifying a 632 bp fragment. A 3 µl sample of the PCR reaction mixture was electrophoresed with a 100 bp DNA ladder on 1% agarose gel to determine the presence and size of the amplified DNA bands, that were visualized by Sybr-Safe staining. A selection of positive PCR’s were purified by standard ethanol precipitation and sent to external facilities for sequencing after purification. DNA sequences were assembled and edited using GENEIOUS and aligned with MAFFT v.6 using the G-INS algorithm and default values for other parameters (Katoh & Toh, 2008). To identify the corresponding supergroup of Wolbachia, we compared our sequences with previously classified sequenced according to Zhou, Rousset & O’Neill (1998), obtained from GenBank.

Results

Combined phylogeny/phylogeography

There were no length differences in the protein coding genes with the exception of Wg, with variation of a single AA in the ingroup. Length variation in the ribosomal genes ranged between 691–695 bp for the ingroup. The best partition scheme included a separation by codon position, but the Bayesian analyses did not converge adequately due to insufficient data to estimate some of the parameters, so we used the second best option, a partition by genes. The best evolutionary models were GTR+I+G for the COI-5′, COI-3′, 16S rRNA+tRNA-Leu and Wingless; GTR+G for nad1, and Trn+I+G for H3.

The ultrametric tree obtained with the combined mitocondrial and nuclear data showed a topology compatible with that obtained in García-Vázquez et al. (2016), with strong support for the monophyly of all studied species groups but poorly resolved relationships within the western clade (Fig. 2). In the D. aubei group we recovered the Pyrenean endemic D. delarouzei as paraphyletic, at the base of the remaining species of the group, which monophyly was strongly supported (Fig. 2). Within the later, the Sicilian and southern Italian D. semirufus were monophyletic and sister to D. aubei plus D. semirufus from the northern Apennines (Tuscany and Emilia-Romagna) and southeastern France. The Pyrenean D. aubei were paraphyletic at the base of a strongly supported clade including first specimens from the Cevennes (geographically closer) and then from the Alps Maritimes, Italian Alps and south Germany (Black Forest), plus the Alpine D. semirufus (Fig. 2). The analysis of the mitochondrial data alone divided the D. aubei group into two clades well separated geographically: one formed by specimens of D. aubei aubei and D. semirufus from the Alps, Black Forest, French Massif Central and northern Appenines, and the other clade subdivided into two sister groups, one formed by specimens of D. delarouzei and D. aubei sanfilippoi from northern Spain and Pyrenees and the other by D. semirufus from Sicily and central-southern Apennines (see Fig. S1). Contrary to the results with the combined or the mitochondrial data alone, the nuclear data recovered a monophyletic D. aubei within a paraphyletic series including the other two species of the group (Fig. S2).

Figure 2 Phylogenetic tree of the studied groups of Deronectes.

Ultrametric tree obtained with BEAST with the combined nuclear and mitochondrial sequence and a partition by gene. The tree had an arbitrary root distance, but the scale is adjusted to the age of the basal node obtained with the mitochondrial tree (Fig. S1). Numbers on nodes, Bayesian posterior probabilities. See Table S1 for details on the specimens and localities. Habitus photograph, D. latus (from Lech Borowiec, http://www.colpolon.biol.uni.wroc.pl/index.htm).

The species of the D. latus group were included in the eastern clade, as sister to the species of the D. parvicollis group. Within the D. latus group, the Italian D. angelinii was sister to the rest, which included the Iberian D. angusi and the widespread D. latus with the Turkish D. toledoi nested within it (Fig. 2). Within the D. platynotus group, the two species of the group (D. platynotus and D. costipennis), and the subespecies D. c. costipennis, were respectively monophyletic and well supported (Fig. 2). In the D. moestus group there was a deep intraspecific divergence in D. moestus. A clade with a predominantly western distribution included the specimens from Morocco, southern Spain and southeastern France as sister to the Mallorcan endemic D. brannani (Fig. 2). Its sister clade, with a predominantly eastern distribution, included specimens of D. moestus from northern Spain to Bulgaria and with specimens of the two previously recognised morphological subspecies (D. m. moestus and D. moestus inconspectus).

All species of the four studied groups were estimated to be of Pleistocene origin with the exception of the deepest clades within the D. moestus complex, dated to the Pliocene-Pleistocene boundary. Intraspecific variation was very recent (from the Upper Pleistocene), as was the separation between some species, which have still not reached monophyly (e.g., within the D. latus or D. aubei groups) (Fig. 2).

Demographic models and detailed phylogeography

In all groups a strict molecular clock was preferred over the relaxed lognormal, which was implemented in all analyses of demographic models (see Table 1 for AICM values). The run with a relaxed lognormal clock in the D. latus group did not converge adequately despite running for 100 million generations, especially for parameters of the 16S gene likely due to insufficient variation. Exploratory analyses of the topology and divergent dates showed no substantial variation with respect to the strict clock model, which was also adopted.

Table 1 Analyses of the best molecular clock model and population coalescent model for each group, including AICM values and standard error (SE).

In bold, best AICM value for each pair. Differences below 2 were not considered as significant. The relaxed model for D. latus failed to converge and was not considered (see ‘Results’).

Group	Clock model	AICM	SE	Coalescence model	AICM	SE	
D. aubei	Relaxed	8235	+∕ − 0.33	Constant	8196	+∕ − 0.27	
	Strict	8196	+∕ − 0.13	Exponential	8199	+∕ − 0.28	
D. latus	Relaxed	–	–	Constant	7274	+∕ − 0.09	
	Strict	7274	+∕ − 0.09	Exponential	7274	+∕ − 0.22	
D. moestus	Relaxed	8734	+∕ − 0.30	Constant	8720	+∕ − 0.25	
	Strict	8720	+∕ − 0.15	Exponential	8716	+∕ − 0.16	
D. platynotus	Relaxed	7052	+∕ − 0.04	Constant	7049	+∕ − 0.07	
	Strict	7049	+∕ − 0.18	Exponential	7046	+∕ − 0.07	

The coalescent analyses with logistic and expansion grow models did not converge adequately in any group. The exponential grow model performed significantly better in the D. platynotus group and the D. moestus complex, and the constant size model in the D. aubei group, while there was no significant difference between both models in the D. latus group (Table 1). The Bayesian skyline plot was very similar for the D. latus and D. aubei groups, with a nearly constant effective population size with a recent, postglacial increase after a slight decline (Fig. 3). In the D. moestus complex it also remained constant until recent times, but with a recent sharp decline followed by a fast increase recovering the previous effective population size. On the contrary, in the D. platynotus group there was a gradual decrease ending in a sharp decline (Fig. 3).

Figure 3 Demographic history of the studied groups of Deronectes.

Bayesian Skyline plots for each species group of the analyses of the mitochondrial sequence, assuming a strict clock (see ‘Material and Methods’ for details). Thin lines, 95% confidence intervals; horizontal axis, time (MY); vertical axis, effective population size (NeT).

Due to the absence of outgroups in the phylogeographic analyses in BEAST the origin of each of the individual clades was reconstructed in a position geographically intermediate between the two basal nodes, and was not interpreted.

In the D. aubei group there were three main lineages geographically well differentiated. The north of the Iberian peninsula (Pyrenees and Cantabrian mountains) was reconstructed as having been populated from east to the west by D. delarouzei and D. a. sanfilippoi; Sicily and the central and southern part of the Italian peninsula by the southern D. semirufus; and the Alps and nearby areas by D. a. aubei and the northern D. semirufus. Within the later, the expansion seems to have been from the Alpes Maritimes and the Italian Piamonte to respectively the Massif Central, the north side of the Alps, including the Black Forest, and the northern Apenines (see Fig. 4 for the phylogenetic tree and Fig. 5 for the map with the phylogeographic reconstruction).

Figure 4 Ultrametric time calibrated tree of the D. aubei group.

Ultrametric time calibrated tree obtained with BEAST with the combined nuclear and mitochondrial sequence of all sampled specimens of the D. aubei group, using the coordinates of each locality as a quantitative trait and the best population coalescent model for each group. Numbers on nodes, Bayesian posterior probabilities. See Fig. 5 for a graphic representation of the reconstructed geographical coordinates, and Table S1 for details on the specimens and localities.

Figure 5 Phylogeographic reconstruction of the history of each lineage.

Phylogeographic reconstruction through time in Google Earth (http://earth.google.com) of the history of each lineage, from the origin until the current distribution, based on the results represented in Figs. 4, 6 and 7. Parabolas represent the reconstructed displacements between nodes in the phylogeny (in red, basal nodes). Yellow arrows mark the general direction of the range expansions, and the white dashed line the hypothesized area of origin of each group. (A) D. aubei group; (B) D. latus group; (C) D. platynotus group; (D) D. moestus complex. Photo credit: Image Landsat, IBCAO and U.S. Geological Survey. Data SIO, NOAA, US Navy, NGA, GEBCO.

The reconstruction of the D. latus group showed an initial expansion into Italy at the origin of D. angelinii. This initial expansion was followed first by the split between the eastern D. toledoi (NE Turkey) and the remaining western lineages, with a second expansion including the colonization of the Iberian peninsula by D. angusi and large areas of central and northern Europe (including the British Islands) by D. latus (Figs. 5 and 6). One of the sequenced specimens of D. toledoi was, however, nested within D. latus (Fig. 6).

Figure 6 Ultrametric time calibrated trees of the D. latus and D. platynotus groups.

Ultrametric time calibrated trees obtained with BEAST with the combined nuclear and mitochondrial sequence of all sampled specimens of (1) the D. latus group and (2) the D. platynotus group, using the coordinates of each locality as a quantitative trait and the best population coalescent model for each group. Numbers on nodes, Bayesian posterior probabilities. See Fig. 5 for a graphic representation of the reconstructed geographical coordinates, and Table S1 for details on the specimens and localities.

In the D. platynotus group the widespread D. platynotus originated in the Balkans, from where expanded eastwards towards the Carpathians and northwards to reach central Europe (Figs. 5 and 6).

The western clade of the D. moestus complex was reconstructed as having an origin between the Balearic islands (with D. brannani) and north Morocco, from were it expanded north to the rest of the Iberian peninsula and southern France. The eastern clade was reconstructed as having an origin between the Tyrrhenian islands and southern Italy, from were one lineage expanded to the north of the Iberian peninsula and north Italy up to Slovenia, and another to Sicily, southern Italy and the Balkan peninsula (Greece and Bulgaria). The islands of Corsica, Elba and Sicily had specimens of different origins within the eastern clade, but the sampled Sardinian haplotypes had a single origin, with a back colonisation to Corsica (Figs. 5 and 7).

Figure 7 Ultrametric time calibrated tree of the D. moestus complex.

Ultrametric time calibrated tree obtained with BEAST with the combined nuclear and mitochondrial sequence of all sampled specimens of the D. moestus complex, using the coordinates of each locality as a quantitative trait and the best population coalescent model for each group. Numbers on nodes, Bayesian posterior probabilities. See Fig. 5 for a graphic representation of the reconstructed geographical coordinates, and Table S1 for details on the specimens and localities.

Presence of Wolbachia

We did not detect Wolbachia in any of the eight tested specimens of the D. platynotus and D. latus groups, with the only exception of one specimen of D. angelinii, positive for the supergroup B of Wolbachia (Table S1). Within the western clade of the D. moestus group we did not identify Wolbachia in any of the six specimens tested from Mallorca, Morocco and southern Spain, but the two tested specimens of D. moestus from Southern France were positive for supergroup B. Of the 15 tested specimens of the eastern clade, three were positive for supergroup A in the Pyrenees and northern Italy and four for supergroup B in Sicily and Greece. The three tested specimens from the central Apenines and Elba were negative (Table S1).

Within the 25 tested specimens of the D. aubei group all were positive, with the only exception of the six specimens of D. a. sanfilippoi from the Cantabrian Mountains, which were negative. All the three tested specimens of D. a. sanfilippoi from the Pyrenees were, on the contrary, infected. All infected populations of the group were positive for supergroup A, with the exception of the two specimens from the French Massif Central and one isolated specimen of D. semirufus from the Abruzzo, positive for supergroup B (Table S1).

Discussion

Recolonization process

The species of Deronectes with a widespread European distribution have all an origin in the southern Mediterranean peninsulas, in agreement with the common pattern of recolonisation after the glacial cycles (Hewitt, 2000). However, there are some fundamental differences with the standard models, as the distribution of the widespread species does not include the southern peninsulas, which are occupied by vicariant taxa of recent origin. Thus, D. latus is replaced in northern Anatolia by D. toledoi, in the Italian peninsula by D. angelini, and in the Iberian Peninsula by D. angusi (Fig. 1). Within D. aubei the northern taxa (D. aubei aubei and the northern clade of D. semirrufus) are restricted to south Germany, the Alps and the Massif Central, being replaced in the north of the Iberian peninsula by D. aubei sanfilippoi and in central and south Italy and Sicily by the southern clade of D. semirrufus. The situation with D. platynotus and D. moestus is more complex, in the former due to the disjoint distribution, and in the later due to unrecognised ancient diversity. However, in the Balkans D. platynotus only reaches the northern mountain chains, in which populations present some morphological differences that may warrant a taxonomic recognition (H Fery, pers. comm., 2016). The southernmost populations of the species, in north Greece, are considered a distinct subspecies with a very restricted distribution, of which no specimens could be obtained for study (D. platynotus mazzoldii, Fery & Brancucci, 1997). There is additional evidence that the species currently restricted to Mediterranean peninsulas were never present in central and northern Europe, as there are no known records of Quaternary fossil remains of any of them (Abellán et al., 2011).

The existence of these two partly non overlapping species pools (Mediterranean species with ranges never extending to northern Europe, and northern species with limited southern distributions) is consistent with increasing evidence from molecular studies of the role of the Mediterranean peninsulas as a source of endemism (see e.g., Bilton et al., 1998; Petit et al., 2003; Ribera & Vogler, 2004). This suggests that after recolonisation during an ecologically optimal period, northern populations become isolated from their source areas. At the end of each Pleistocene Glacial cycle environmental conditions in the newly deglaciated areas in central and north Europe could have been optimal for running water organisms typical of mountain streams, such as Deronectes. The likely vast number of small streams from the thaw of the ice sheet, even in lowland areas, could have represented an ecological opportunity for these species, favouring range expansions to new empty areas without developed communities but with homogeneous favourable conditions (Ribera et al., 2011). When ecological conditions changed populations become increasingly isolated, resulting in the currently recognised taxa.

However, only some species in these lineages of poor dispersers expanded their ranges, while the rest remained confined to restricted areas in the southern Peninsulas. The species with widespread distributions do not seem to share any particular phylogenetic pattern. They are not the oldest species in their clades, which could have lead to the hypothesis that their wider ranges were due to a longer time to disperse (Willis, 1926; Gaston, 2003). On the contrary, they are in most cases nested within southern species or populations. Another possibility is that their physiological or ecological tolerances favoured their range expansion. There is some evidence that differences in the ecological tolerance of the species are related to their geographic range extent (Addo-Bediako, Chown & Gaston, 2000; Gaston & Spicer, 2001; Gaston, 2009). Thus, widespread species would have a higher ecological plasticity while species with restricted distributions may have a limited physiological tolerance (West-Eberhard, 2003). Among the many possible ecological or physiological factors, thermal tolerance have been frequently linked with distributional ranges (e.g., Stillman, 2002; Somero, 2005; Verdú & Lobo, 2008). Previous physiological studies with Deronectes showed than widespread, more northerly distributed species have broader thermal tolerance than their restricted southern relatives (Calosi, Bilton & Spicer, 2008; Calosi et al., 2010). Thus, the most widespread species of the genus, D. latus, had also the greatest thermal window (Calosi et al., 2010). However, there is a poor adjustment of experimental data with ecological data derived from the localities they currently occupy (Sánchez-Fernández et al., 2012). With the available information (with no data on intraspecific variability) it is also not possible to assess if this wider thermal tolerance was previous to the range expansion, making it possible, or if it was developed after the range expansion, as a consequence of being exposed to a wider range of climatic conditions.

An alternative hypothesis is that the widespread species did not have any particular physiological or ecological character favouring their range expansion, but just took advantage of a privileged geographical position. They may have been the ones that happened to be at the edge of the newly deglaciated areas. According to our results, populations in mountain ranges at the northern edge of the southern peninsulas played a key role in the recolonization of glaciated areas. Thus, the D. latus group is the westernmost lineage within the eastern Deronectes clade, and the central European populations of D. platynotus appear nested within those of the northern Balkans. Within the Iberian Peninsula, with numerous species of Deronectes with a restricted distribution in most of its mountain systems (Millán et al., 2014; García-Vázquez et al., 2016), the D. aubei group is restricted to the northernmost ranges (Cantabrian mountains and the Pyrenean area, Fig. 1). As already noted, the situation within the D. moestus group is more complex, due to the deep divergences between some lineages. Under this hypothesis, historical more than intrinsic factors would determine which species become widespread and may be the origin of further diversification (Ribera et al., 2011). The evidence supporting the role of geographic position in facilitating range expansions relies on the assumption that southern species have maintained their geographic ranges through the last glacial cycles, so that it is possible to infer their past distribution from their current location. Although this assumption has been challenged (e.g., Gaston, 2003; Losos & Glor, 2003), it has been repeatedly shown that in poorly dispersing lineages range movements do not erase completely the geographic signal from the past (Barraclough & Vogler, 2000; Abellán & Ribera, 2011; Ribera et al., 2011).

Demographic history of the recolonization

The dynamics of the range expansions was not the same in all the studied groups, as shown with the reconstructions of the Bayesian skyline plots (Fig. 3). Thus, while in the D. aubei and D. latus group we estimated a fast expansion through western Europe (D. aubei) or the whole continent (D. latus), leading to the current continuous distributions, in the D. platynotus group there was a sharp decline in the effective population size, which may reflect their current discontinuous distribution, with isolated populations in the east (D. platynotus) and the west (D. costipennis). More difficult to interpret is the demographic history of D. moestus, with a sharp decline corresponding roughly to the last Glacial Maximum (LGM) and a subsequent expansion. In this complex, the late Pleistocene climatic cycles likely resulted in the isolation of the different clades recovered in the phylogeny, some of which experienced subsequent expansions when conditions improved after the LGM.

Wolbachia infections

We found a high prevalence of Wolbachia infection among Deronectes, with specimens infected in five of the eleven studied species (45%). This percentage is concordant with recent estimations of a 40% of Wolbachia prevalence among arthropods (Zug & Hammerstein, 2012) and 31% in a group of families of aquatic beetles, among them Dytiscidae (Sontowski et al., 2015), thus making it the most successful endosymbiont on earth. As expected, all infected specimens were for supergroups A and B, the most common in arthropods (Werren, Baldo & Clark, 2008).

The prevalence of Wolbachia was particularly high in the D. aubei group, in which all sequenced populations were infected with the only exception of the Cantabrian Mountains, the westernmost area of distribution of the group. It is interesting that this group had also a marked contrast between the results of the mitochondrial and nuclear phylogenetic reconstructions, although with the current data it is not possible to establish a clear link between the Wolbachia infection and this discordance. Previous results, with a limited sampling, reported two mitochondrial groups east and west of the Rhone river (García-Vázquez et al., 2016). This scenario was modified with the inclusion of additional samples from the margins of the distribution of the group, with the populations of D. semirrufus from Sicily and the central and southern Apennines joining the Iberian and Pyrenean clade (including D. aubei sanfilippoi and D. delarouzei), and the populations of D. aubei from the French Massif Central joining the Alpine and German Black Forest clade (including D. semirrufus and D. aubei aubei). Thus, the mitochondrial data separated the D. aubei group in two main lineages north and south of latitude 44°N. On the contrary, nuclear data largely recovered monophyletic species, defined according to their external morphology and the male aedeagus (Fery & Brancucci, 1997). In the mitochondrial tree, the two subspecies of D. aubei as defined with morphology were recovered in different clades, with the only exception of the populations of the French Massif Central and the Cevennes (west of the River Rhone), morphologically closer to the Pyrenean D. aubei sanfilippoi (Fery & Brancucci, 1997; H Fery, pers. comm., 2016) but grouped with D. aubei aubei from the Alps and south Germany. In the evolutionary history of the D. aubei group, there seem thus to be a first event leading to mitochondrial introgression and geographic isolation within D. aubei and D. semirrufus, separating in the former the recognised subspecies D. aubei sanfilippoi and D. aubei aubei, and in the later the populations from Sicily and central and Southern Italy and those from the Alps, which although not currently recognised as distinct taxa they have diagnostic morphological differences (D García-Vázquez & I. Ribera, 2016, unpublished observations and H Fery, pers. comm., 2016). Subsequent to this, there may have been a secondary event of introgression between the two subspecies of D. aubei in the populations of the Massif Central and the Cevennes. These occupy a geographically intermediate position between the western D. aubei sanfilippoi and the eastern D. aubei aubei, with the general morphology of the former but a mitochondrial genome related to the later.

Supplemental Information

Supplemental Information 1 Supplemental Information including all raw data (Accession Numbers and geographical coordinates of the specimens studied)

Click here for additional data file.

We especially thank all collectors mentioned in Table S1 for their invaluable help in providing material for the study, and R Alonso and A Izquierdo for laboratory work. We also thank DT Bilton and LF Valladares for their help with the study of Deronectes.

Additional Information and Declarations

Competing Interests

Author Contributions

DNA Deposition

Data Availability

The authors declare there are no competing interests.

David García-Vázquez conceived and designed the experiments, performed the experiments, analyzed the data, contributed reagents/materials/analysis tools, wrote the paper, prepared figures and/or tables, reviewed drafts of the paper.

Ignacio Ribera conceived and designed the experiments, analyzed the data, contributed reagents/materials/analysis tools, wrote the paper, prepared figures and/or tables, reviewed drafts of the paper.

The following information was supplied regarding the deposition of DNA sequences:

Sequences were submitted to the EMBL database and are accessible in GenBank with accession numbers LT601818–LT602609 (sequences of specimens of Deronectes) and LT602610–LT602633 (sequences of the Wolbachia endoparasites).

The following information was supplied regarding data availability:

The raw data consists of the sequences (which have been submitted to GenBank) and the geographical coordinates of the localities in which the species were found, which are provided in Table S1.

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
