# Peer review of "The origin of widespread species in a poor dispersing lineage (diving beetle genus Deronectes)"

_PeerJ, doi:10.7717/peerj.2514_

## Round 0.1 · original submission · Minor Revisions

Both reviewers agree in that the ms deserves being published after minor revisions. Please try to give answer to the two reviewers comments, specially pay attention to reviewer-2 proposal of performing path sampling to compare with the results of AICM.

Best wishes,
Marta

Reviewer 1 ·

Basic reporting

The authors of the present paper entitled "The origin of widespread species in a poor dispersing lineage (diving beetle genus Deronectes)" have obtained an expanded time-calibrated phylogeny on the basis of that of García-Vázquez and collaborators (2016) using more specimens and genes but focusing on those species with a wide European distribution range (i.e. aubei, moestus, latus, platynotus). The aim of the authors is to infer the different dispersal routes and times that could explain the present distribution range of the four lineages. Additionally, the authors have looked for the presence of the parasite Wolbachia in these lineages. It is thought that Wolbachia infections are the cause of divergences between the mitochondrial and nuclear phylogenies. I would like to point here that I am not an expert on beetles so the different aspects of the biology of this group are outside the scope of my knowledge.

The paper is generally well carried out and I would like just to point out minor corrections:

> It seems that there is something wrong with the figure numbers/figures shown/figures mentioned in the manuscript. The legend of Figure 4 says "Ultrametric calibrated trees obtained with BEAST with the combined nuclear and mitochondrial sequence of all sampled specimens of the four studied lineages, using the coordinates...(1) D. aubei group; (2) D. moestus complex; (3) D. latus groups; (4) D. platynotus group.". Actually, these are presented as figures 4, 5, 6 (A?) and 6 (B?) respectively. Figure 4 legend also says "See Fig. 5 for a graphic representation of the reconstructed geographical coordinates...". I understand it might be referring to figure 7 (I will refer to this figure as figure 5/7 from now on).

> Figure 5/7 looks a bit rough. I suggest to shade/mark the hypothesized area of origin of the different groups. Yellow lines could be thicker. I also suggest to remove the Google Earth letters from the figures.

> Please, explain in the figure 5/7 legend what do the black and red parabolas mean.

> I suggest to start the sentence "The reconstruction of the D. latus group..." in line 324 in a new paragraph, so every group is discussed in a different paragraph. On the other hand, I would show these results in the same order than the tree figures (i.e. (1)aubeis, (2)moestus, (3)latus, (4)platynotus). The same would apply to figure 5/7 (change of the order of the maps).

> The distribution of D. moestus complex in the figure 1 of the present paper and that of Garcia-Vázquez et al., 2016 does not match completely. D. moestus complex is disributed in Morocco, Algeria and Tunisia in Africa according to the latter paper but confined to Morocco according to the present work.

> Line 186: Please mention to which external facilities the purifications were sent.

> Line 430: norhtern -> northern

Experimental design

No comments.

Validity of the findings

No comments.

·

Basic reporting

The manuscript by Garcia-Vazquz and Ribera (#11288) shed light on the diversification of a particular lineage of diving beetles of the genus Deronectes, D. modestus group plus D. brannani, from Western Mediterranean basin. The use a dense sampling (297 specimens from 109 populations) and molecular phylogeograpic and coalescent tools to estimate the systematics and the tempo and mode of diversification of the sampled populations of this group. They estimated their origin of D. modestus group plus D. brannani lineage during the Pleistocene. The geographic pattern found seems to be driven by the glacial cycles which after global warming allowed to expand their ranges to central and northern Europe by colonizing the new optimal ecological conditions. Afterwards, some population become isolated and formed new species with narrow ranges. Moreover, they test whether presence of Wolbachia infection could play an important role of this diversification pattern. This is a neat phylo-bio-geographic study which test several diversification hypothesis to explain why a linage show both narrow and wide geographic ranges. Finally, the manuscript is well written and eady to follow, and Tables and Figures relevant. My recommendation is to accept this manuscript for publication in PeerJ after minor changes are made.

Experimental design

The experimental design is correct and detailed. Topic is also within the scope of the Journal. However, since differences among clock models and also among coalescent models are extremely narrow I suggest to compare them using path-sampling which is more accurate than AICM (Baele et al 2012, Mol. Biol. Evol. 29, 2157–67). Under some circumstances, path-sampling analyses do not converge due to implementing inappropriate priors. One can overcome the lack of convergence by implementing narrower prior values in a log-normal distribution. In fact, the presence of a very large value in Table 1 for AICM, i.e. relaxed clock in D. latus (37123), suggest that that analysis got trapped in a local minimum (or it is a typo). Larger running times with large burnin could also improve convergence and obtain ESS values > 200 for most parameters.

Validity of the findings

Findings reported here were validated using the mos recent and accurate statistical tests based on molecular data and performed under maximum likelihood and Bayesian analyses.

Additional comments

4.1 The meaning of the words “lotic” and “lentic” are introduced early in the manuscript. Since both words are very specific and PeerJ is intended for a broad audience, I thing it would be easier for readers the use of expressions running and standing water throughout the text.
4.2 The introduction includes a very detailed hypothesis about diversification that perhaps could be discussed further in the discussion section.
4.3 The length of the section taxonomic background could be reduced, particularly the detailed explanation about Deronectes which are not the focus of the present study, i.e. D. modestus group plus D. brannani. A brief summary with 2-3 sentences and the map are more than enough for the background.
4.4 Fig. 2. Use white spots rather than black ones since background is already very dark.
4.5 Since Wolbachia infection seems to not play a key role on the diversification, I suggest to remove all comment about Wolbachia from the manuscript.

---

## Round 0.2 · accepted · Accept

The two reviewers agree in that their comments and corrections have been adequately addressed by the authors in the new version, so the ms is now ready to be accepted.

Reviewer 1 ·

Basic reporting

No Comments

Experimental design

No Comments

Validity of the findings

No Comments

·

Basic reporting

The revised version of the manuscript fixed all minor errors and reply to all my questions. Hence i suggest its publication in the present form.

Experimental design

Correct

Validity of the findings

Correct

Additional comments

None